# Development and Integration of Patient-Reported Measures into E-Health System: Pilot Feasibility Study

**DOI:** 10.3390/healthcare11162290

**Published:** 2023-08-14

**Authors:** Goda Elizabeta Vaitkevičienė, Karolis Ažukaitis, Augustina Jankauskienė, Justė Petrėnė, Roma Puronaitė, Justas Trinkūnas, Danguolė Jankauskienė

**Affiliations:** 1Center of Pediatric Oncology and Hematology, Vilnius University Hospital Santaros Klinikos, 08406 Vilnius, Lithuania; 2Institute of Clinical Medicine, Faculty of Medicine, Vilnius University, 03101 Vilnius, Lithuania; karolis.azukaitis@santa.lt (K.A.); augustina.jankauskiene@santa.lt (A.J.); roma.puronaite@santa.lt (R.P.); 3Center of Pediatrics, Vilnius University Hospital Santaros Klinikos, 08406 Vilnius, Lithuania; 4Center of Hematology, Oncology and Transfusiology, Vilnius University Hospital Santaros Klinikos, 08406 Vilnius, Lithuania; juste.petrene@santa.lt; 5Center of Informatics and Development, Vilnius University Hospital Santaros Klinikos, 08406 Vilnius, Lithuania; justas.trinkunas@santa.lt; 6Faculty of Fundamental Sciencies, Vilnius Gediminas Technical University, 10223 Vilnius, Lithuania; 7Health Research Laboratory, Mykolas Romeris University, 08303 Vilnius, Lithuania; djank@mruni.eu

**Keywords:** patient-reported outcome measures, patient-reported experience measures, patient involvement, electronic health records, patient-centered care, self-report, kidney disease, hematological disease

## Abstract

Patient-centered care is recognized as a key element in recent healthcare management strategies. However, the integrated collection of patient feedback capturing the entire journey of patients with complex medical conditions remains understudied. Herein, we aimed to describe the development of an instrument prototype for the collection of PROMs and PREMs that would encompass a whole patient journey at a single time point. We further describe the process of its integration into a hospital’s information system (HIS) and the results of a pilot feasibility study in adult patients with kidney and hematological diseases. We developed an instrument consisting of original PREM and generic EQ-5D-5L questionnaires. E-questionnaires were handled with REDCap software (version 12.5.14) and integrated into the HIS. Patients refusing to use e-questionnaires (48%) were offered paper administration and were older (64 vs. 50 years). The overall response rate for e-questionnaires was 57.1% with a median completion time of 2.0 and 3.7 min for PROM and PREM, respectively. Psychological and social services and primary care setting (diagnosis establishment and involvement in continuous care) were identified as most problematic. The majority of PREM dimensions encompassing different levels of care significantly correlated with PROM responses. Our data indicate the feasibility and potential relevance of the proposed approach, although wider-scale studies in diverse settings are needed.

## 1. Introduction

Patient engagement and patient-centered care are considered as key elements in recent healthcare quality management strategies [1,2]. Thus, improving the quality of healthcare services by focusing on the patient’s needs is becoming increasingly relevant in the healthcare systems worldwide. The rapid advances in the information technology sector and health digitalization led to the intensifying use of digital tools for patient-centered care that showed to be more cost-efficient [3] and less time-consuming, as well as associated with higher patient satisfaction [4,5] and improved patient–physician communication [6]. On the other hand, digital approaches to collecting patient feedback may face challenges, such as limited response rate or omitting patients who lack computer literacy. The integration of patient-reported measures’ collection into the digital services is not yet a routine practice in many institutions. This is largely due to difficulties in finding the most appropriate models for implementation and contradictory evidence related to the value of such strategies [7,8].

Patient-reported outcome (PROM) and experience (PREM) measures are most commonly used patient feedback instruments, making it possible to gather information about subjective health outcomes and patient perspectives on what happened during the healthcare provision process. That, among other purposes, can also serve for the assessment of healthcare service quality. PROMs and PREMs are standardized or custom-developed questionnaires or scales used to collect patient-reported health-related outcomes and experiences within a healthcare system. The major advantage of both PROMs and PREMs is their reliance on information received directly from the user of the healthcare system, i.e., the patient, without interpretation from healthcare providers [9]. Furthermore, some data suggest a bidirectional interaction between patient experiences and outcomes, i.e., health-related outcomes can affect a patient’s perceptions of healthcare services, while the latter, to a slightly lower extent, can also impact subjective health outcomes [10,11]. However, PROMs and PREMs have also received criticism concerning their accuracy and reliability to be implemented in routine care compared to objective clinical measures (e.g., blood pressure) [12,13].

Although PROMs are frequently employed for individual patient management, both PROMs and PREMs can also be used in a wider scope, i.e., meso (institutional) or macro (healthcare system) levels. In the latter cases, these instruments may serve as important indicators of healthcare service quality from the patient perspective. As such, they may be useful for healthcare quality improvement, particularly when implementing patient-centered care or performing benchmarking [14]. PREMs are most frequently used at micro or meso levels, but their use at the macro level, particularly aiming to capture the performance of the healthcare system and its different levels, is limited. Moreover, there remains a lack of evidence to defend that the use of patient-reported measures at meso or macro levels positively influences micro-level performance [15].

PREMs also typically capture patient experiences for a single encounter with the health system in a particular health context. When tracked longitudinally, they are able to show trends over time and provide an opportunity to monitor the effects of healthcare quality improvement strategies [16]. Although likely providing the greatest accuracy, this may also pose several challenges if employed at the level of a healthcare system, particularly that related to the requirement of different instruments for different settings and a large amount of data produced. On the other hand, in the modern healthcare system, when prioritizing a holistic approach, the tracking of patient experience throughout the entire patient journey is of particular importance [17].

Thus, to evaluate the overall experience in the healthcare system, especially for the patients with chronic diseases or complex medical conditions requiring frequent interactions with healthcare providers, it is important to extend the scope of self-reported measures to the patient’s whole journey. This may include the assessment of experiences during encounters with healthcare providers at primary care, secondary and tertiary care levels, as well as rehabilitation or social services. It is known that prior experiences have an impact on a patient’s attitude and perspectives for the remaining journey in the healthcare system [18,19,20]. Understanding the experience at different stages of patient journey may provide a fuller picture on how the healthcare system functions, as well as how these different experiences interact with each other and with health-related outcomes.

Considering the aforementioned needs and challenges, we aimed to develop and pilot an instrument prototype targeting patients with chronic and complex conditions that would capture both PROMs and PREMs throughout an entire patient journey at a single-time point. Being one of the largest tertiary care institutions in the Baltic region, we chose kidney and hematological diseases as representative populations of patients undergoing complex care pathways. Serving as the ultimate point of care for such patients provides a unique opportunity to evaluate the functioning of the overall healthcare system. In addition, we developed and tested a process for integrating this instrument into a hospital’s information and quality management system. Following a successful implementation at the healthcare provider level, the instrument and its implementation process are expected to serve as a prototype for nationwide integration into the national e-Health system. This may provide an opportunity for the regular monitoring of overall healthcare system performance and benchmarking as well as for the further adaptation of different conditions and settings.

Primary aim. The overall objective of this study was to describe the development of an instrument (including contents, process and integration) for PROMs and PREMs collection, capturing the whole patient journey at a single time point that could be integrated into the hospital electronic medical records system (EMR).

Secondary aim. To test the feasibility of the instrument with the representative group of patients that require long-term management (chronic kidney and hematologic diseases) by the specialists in all three levels of the national (Lithuanian) healthcare system.

## 2. Materials and Methods

### 2.1. Setting and Study Population

A cross-sectional study involving patients requiring long-term medical care at all three healthcare levels was performed. Patients with hematological diseases or chronic kidney diseases (CKDs) were selected as representative populations. Inclusion criteria were as follows: adult patients (≥18 years of age) with the diseases diagnosed ≥6 months ago and either (1) receiving chemotherapy or immunotherapy on an outpatient or inpatient bases, recipients of hematopoietic stem cell transplantation, treatment for CKD, receiving kidney replacement therapy or recipients of kidney transplant or (2) undergoing follow-up with ≤12 months after the end of active treatment. This study was undertaken in two centers: (1) the center for Hematology, Oncology and Transfusiology and (2) Nephrology center at Vilnius University Hospital Santaros Klinikos (VUH SK), Vilnius, Lithuania during the period from October 2022 to March 2023. All consecutive patients meeting eligibility criteria and seen in the day-care, inpatient and outpatient clinics were invited to participate in the study.

### 2.2. Ethics Approval

The study was conducted according to the guidelines of the Declaration of Helsinki and approved by the Regional Biomedical Research Ethics Committee of Vilnius, Lithuania (number 2022/9-1465-932, date of approval 23 September 2022). Patient consent was waived as anonymized data were used for analysis, and anonymization was performed by automatic measures wherein none of the patients could be identified by neither of the investigators.

### 2.3. Methods

The patients at their outpatient, day care visit or during their in-patient treatment phase were invited to complete the survey by responding to an electronic questionnaire. The patients were asked to provide an e-mail address and received a unique link to the questionnaires. Further details on the integration of this process into the hospital’s information system (HIS) are described in the following section. E-questionnaires could be filled by using any electronic equipment, including personal computers, tablets or mobile phones. The questionnaires were filled only once by each patient.

For patients who declined to fill out the electronic questionnaire (due to not having access to appropriate devices or other reasons), paper administration versions for both PROM and PREM forms was offered. Responses from paper forms were collected and transferred into the RedCap system by a dedicated data manager.

Completion time (time from first responded question to submission) and completeness rates (proportion of fully completed questionnaires) were only assessed for electronic questionnaires. To compare the instrument feasibility for the patients of different ages, the patients were divided into different age groups: 18–34, 35–44, 45–54, 55–64 and 65+ years.

### 2.4. Instrument Integration into the Hospital Information System

VUH SK operates its own in-house developed integrated hospital information system SANTA-HIS that integrates all services provided by the hospital, including electronic health records (EHRs), Laboratory system, Picture Archiving and Communication System (PACS), the Patient monitoring system register (PMSR), Biobank system, Quality management system, Adverse events management and Power BI for data analytics (Figure 1). SANTA-HIS is based on a repository database and is integrated into the Lithuanian national health information system (Electronic Health Services and Cooperation Infrastructure Information System, ESPBI IS). To implement the solution needed for this project, REDCap system, which is dedicated for any surveys or clinical trials data collection (https://www.project-redcap.org, accessed on 17 May 2022), was selected and integrated into the existing hospital information infrastructure. The e-mail addresses were included by the data manager to the Patient Monitoring System Register (PMSR) in the Electronic Medical Records (abbreviated: ELI) system at the VUH SK. Unique electronic links with a digital identifier to e-questionnaires were sent to provided e-mail addresses via the REDCap system.

The data from completed questionnaires were automatically transferred to the VUH SK data analytics system Microsoft Power BI Report Server (Power BI) [21]. The data of each completed questionnaire were linked to the unique digital identifier, and the process was fully automated. Patient’s age and gender were automatically retrieved from the hospital information system. All the data in VUH SK’s data analytics system Power BI were systematized, aggregated and analytical models were prepared to evaluate the possibilities of analyzing PROMs and PREMs using a prototype instrument integrated into VUH SK information systems. The developed analytical models did not contain patient-level data and included only aggregated e-questionnaire responses. The architecture of the implemented system is shown in Figure 1.

### 2.5. Instruments Used for Patient-Reported Measures

PROMs were assessed using a standardized generic EQ-5D-5L (EuroQoL Five Dimensions) instrument that consists of five dimensions for self-assessment: mobility, self-care, usual activities, pain/discomfort and anxiety/depression. Outcomes were assessed on a 5-point Likert scale and on a 100-point visual analogue scale (VAS-100) for subjective overall health status estimation [10,22].

An original PREM instrument consisting of 12 questions was developed by the study group following the methodology. First, the study group mapped patient journey starting from the first disease symptoms and the contact of the primary care by visiting the family doctor, through the diagnostic procedures at primary and secondary care levels, to the final diagnosis and treatment at a tertiary center and finally, rehabilitation, psychological support and social services. Second, a focus group involving 12 patients with kidney and hematological diseases that had already passed through all the steps of the patient journey was conducted, aiming to identify the challenges and experiences that the patients encountered at each step. An unstructured interview based on the targeted group technique [23] was used during the focus group study and the recording was transcribed verbatim. Content analysis was performed using the NVivo software, v. 11 by identifying categories and subcategories according to seven categorizing principles [24]. Following these steps, the study group consisting of healthcare quality researchers and healthcare providers drafted a questionnaire aiming to address the issues that were identified in the qualitative analysis of the focus group as most relevant. Finally, a consensus workshop involving all the primary stakeholders, including healthcare providers, patients’ representatives, healthcare managers and policy-makers was organized to revise the questionnaire. The contents of the developed PREM questionnaire are provided as Appendix A. The PREM questionnaire consisted of 12 questions with Likert scale responses ranging from 1 to 5 points (1—fully agree, 5—fully disagree). An option to indicate that the patient did not receive the specified services was provided. Each response was assigned with a value between 0 and 1 with a higher value indicating worse evaluation, and the total PREM score was calculated as the mean of recorded responses. Cases wherein the patients indicated they did not use the specific service were omitted. If more than 50% of the questionnaire values were omitted for the patient, the total PREM score was not calculated.

### 2.6. Statistical Analysis

Descriptive statistics were used for data analysis. Continuous variables were summarized using median and interquartile ranges, and categorical data were expressed as frequency and percentage. The Internal consistency of PROM and PREM questionnaires was assessed by calculating Cronbach’s alpha. Correlation between responses (within and between PROM and PREM items and total of dimension scores) was assessed by calculating Spearman’s correlation coefficients. Pearson’s chi-squared or Fisher’s exact tests were used for categorical and Wilcoxon rank sum or Kruskal–Wallis tests for continuous variables, respectively. R software (Version 4.2.1) [25] and its packages REDCapR [26], eq5d [27], PROscorerTools [28], likert [29], DescTools [30], ggplot2 [31] and Hmisc [32] were used for analysis.

## 3. Results

A total of three hundred and thirty-seven patients were enrolled, including those who provided their e-mail addresses to receive the questionnaire (N = 175) or preferred paper administration (N = 162). The median age of the patients was 58 years and 188 (55.8%) were male. A higher proportion of the patients were treated for hematological illnesses (60.2%). The patients who preferred paper administration were significantly older compared to the patients who agreed to complete e-questionnaires (median age 64 and 50 years, respectively (*p* < 0.001)) (Table 1).

Response rate. Overall response rate (proportion of the patients completing at least one PROM or PREM questionnaire) for the patients completing electronic forms was 57.1% (N = 100) resulting in 179 completed individual questionnaires (representing 48.6% and 53.7% for PROM and PREM, respectively). Both PROM and PREM questionnaires were filled by 79/100 patients (79%). Six patients completed PROM and 15 patients PREM only (3.4% and 8.6%, respectively). The response rates did not differ by sex or age group.

Completion time. The median (IQR) completion time per electronic questionnaire was 2.0 (1.3–2.8) minutes for PROM and 3.7 (2.6–5.3) minutes for PREM. The completion time of the e-questionnaire did not differ by sex, but there was a significant correlation between age and completion time for both PROM (r = 0.231, *p* = 0.041) and PREM (r = 0.414, *p* < 0.001), with younger patients requiring less time to complete the questionnaires (Table 2).

Completeness rate. Out of 85 electronic PROM questionnaires, 69 were fully completed (81.2%), whereas 16 questionnaires were missing a VAS score (18.2%). Electronic PREM questionnaires were fully filled by all 94 patients (100%).

PROM questionnaire results. Of all five PROM dimensions, scores for pain/discomfort were reported as the worst with over one-third (34.7%, N = 86) of the patients experiencing extreme, severe or moderate pain. This was followed by complaints in usual activities, mobility, anxiety/depression and self-care with 31.4% (N = 79), 29.1% (N = 70), 26.4% (N = 65) and 17.6% (N = 44) of patients reporting moderate-to-extreme complaints within these domains (Figure 2A). The median total score was 0.9 (0.8–1), with values ranging from −0.21 to 1, while the median VAS score was 60 (50–80) with values ranging from 5 to 100. Younger patients reported statistically and significantly better scores for mobility (*p* = 0.002) and usual activities (*p* = 0.001) as well as VAS (*p* = 0.016) (Figure 3A,B). No significant differences between male and female respondents were documented (Figure 2B). There was a significant correlation between age and EQ-5D-5L total score (r = −0.289, *p* < 0.001) and VAS (r = −0.202, *p* = 0.002), with younger patients reporting better health quality and better self-care.

PREM questionnaire results. Patient-reported experiences revealed the worst scores for services provided by psychologist or social workers with 25.8% (N = 66) and 28.5% (N = 73) of the patients indicating they disagree or strongly disagree that the psychologist’s or social worker’s program was provided properly (Figure 4A, questions 9 and 10, respectively). This was followed by the issues encountered in the primary care setting with 16.4% (N = 42) and 16.8% (N = 43) of the patients reporting that they disagreed or strongly disagreed that the family doctor was involved in the treatment of the disease or the process from the onset of symptoms to diagnoses was smooth (Figure 4A, questions 1 and 2, respectively). Meanwhile, the best evaluation was dedicated to the tertiary care with more than half of the patients (53.9%, N = 138) strongly agreeing that the treating center environment was patient-friendly and responding to patient’s needs (Figure 4A, question 8). The median total score was 0.3 (0.1–0.4) with values ranging from 0 to 0.7. No statistically significant differences were found when comparing questionnaires by sex (Figure 4B).

A higher proportion of younger patients did not attend the outpatient clinic, whereas older patients visited the outpatient clinic more often as well as perceived the surroundings and the staff of the outpatient clinic more positively (Figure 5, question 4). Older patients rated their experience in the day clinic more positively compared to younger patients (Figure 5, question 5). The younger population (18–44 years) more often disagreed or strongly disagreed that the family doctor was familiar and involved in the patient’s treatment as well as with the statement that the process from the onset of symptoms to disease diagnosis was smooth (Figure 5, questions 1 and 2, respectively).

Both PROM and PREM questionnaires achieved a relatively high Cronbach alpha (PROM 0.869, PREM 0.886), indicating good internal consistency (Table 3). All questions in the PROM questionnaire statistically and significantly correlated with each other and with the overall score (Appendix A). Most PREM questions correlated with each other, and all 12 questions correlated with the total score. Most pairs of the questions of the PROM and PREM were statistically significantly correlated. The overall quality of life score (EQ-5D-5L) statistically significantly correlated with all except one (psychologist’s service) of the PREM dimensions, and the PREM overall score was statistically and significantly correlated with all dimensions of the PROMs questionnaire, including the PROMs’ overall score itself (EQ-5D) (Appendix A).

Power BI was used for data visualization. The visualization model consisted of the plot on demographic characteristics, providing an overview on the number of participants, distribution of the age groups, the numbers of questionnaires that were sent and completed, the average response time and time from opening the questionnaire to completing the questionnaire. Filters were introduced to filter participants by questionnaire type or age group. An example of the output that was available to physicians and healthcare managers within the hospital’s information system is provided in Figure 6.

## 4. Discussion

In the present paper, we describe the process of developing an instrument for the integrated collection of PREMs and PROMs at a meso level (i.e., healthcare service provider level) with successful integration into the HIS. It is expected to serve as a prototype for further development and integration of similar instruments to be integrated into the e-Health system at the national level. The results of our pilot study indicate that the implementation of such an instrument is able to capture different levels of patient journey at a single time point and is feasible at a provider level in the population of adult patients requiring long-term management. However, several challenges for successful functioning were identified. First, the electronic collection of the measures may not be preferred among certain groups of patients, particularly those of older age. The latter group provided different patient-reported outcomes and experiences, and could thus not be omitted. Second, the response rates to the e-questionnaires were relatively low.

However, our study made it possible to identify the most problematic areas through the perspective of patient experiences, namely psychological and social support and performance of primary care sector including the diagnostic process and further continuous patient management. We also observed significant correlations between different PREM questions related to different stages of patient journey and current subjectively reported health status. This provides indirect evidence about the relevance of collecting retrospective patient experiences during their patient journey.

Subsequently, if adopted at the healthcare system level, such an approach could be useful for the benchmarking of healthcare services. The unique aspects of our integrated tool are: (1) capturing of entire patient journey at a single time point as opposed to measuring a single healthcare system encounter, and (2) being able to demonstrate the correlation of reported experiences and patient-reported outcomes at the time of measurement. As such, the instrument fulfils the goal indicated by Davies et al. “to evaluate continuity of care across health services and regions” [33].

Capturing the direct patient feedback has been shown to improve communication between patients and clinicians and stimulate patient engagement in the treatment process [34,35,36]. One of the prerequisites for the effective use of patient-reported measures is the need to collect them regularly as well as the requirement for high response rates. However, the construction of an attractive and user-friendly tool that would make it possible to cover the majority of healthcare system is challenging [13,34,37]. Thus, the design and implementation of patient-reported tools require the involvement of a multidisciplinary team that exhibit necessary competences to ensure the quality and safety of data collection. This is of particular importance when developing tools for capturing the most relevant PROM and PREM questions that maintain clinical value and at the same time can be used with minimum burden to the users [2]. An approach consisting of planning, selection and engagement has been introduced for the development of PREM [38].

We aimed to create a custom-designed PREM tool which could be further tailored to more specific needs of target populations. Aiming to capture the entire patient journey, for the pilot study, we chose patients with kidney and hematological diseases as a representative population that would require long-term clinical management. For the development of our tool, we employed the focus group technique, involving patients with frequent encounters at different levels of the healthcare system to identify their positive and negative experiences, multidisciplinary team of experts and finally a consensus workshop involving all stakeholders to identify the most important and most concerning dimensions, as well as to discuss the accuracy and relevance of the created tool.

Benchmarking is a valuable approach that can be used to optimize patient journey and improve healthcare provision strategies [35,36]. However, there are only a few examples for the use of patient-reported measures by the stakeholders for these aims [39]. We managed to capture and identify specific elements of the patient journey as potential objects for improvement at the national healthcare system. The most unfavorable experiences by our study population were related to primary care, particularly in terms of patient-friendly atmosphere within the clinics.

The digital approach to collecting patient-reported measures and integrating them into the E-systems is still an innovative form of patient engagement. A recent large meta-analysis of patient-reported outcome measures implementation in research and clinical use revealed that, in most of the cases, paper forms were still used with only a very small part collected purely electronically or combining both methods [39]. Age-related digital illiteracy and, in some cases, access to the Internet, had been highlighted by the studies [40]. In the study of geriatric patients, at least 50% of patients were not able to complete digital forms without assistance, with less assistance required for paper forms [41,42].

The primary metric we were interested in was response rate and time taken to complete the form. A relatively low response rate was the challenge we faced. Only about half of the patients agreed to provide their e-mails to receive the link and completed the questionnaires. However, similar challenges in response rates were reported by other studies aiming to implement the systems for capturing patient-reported outcomes. The mode of administration is of importance. The review study by Wang et al. revealed that electronic-only format patient reports have the lowest response rates (RRs), at around 42%, compared to mixed paper and electronic (RRs around 70%) or registry-based (RRs > 90%) questionnaires [43]. The studies show that younger age, having a native-speaking language background and even having a longer waiting time when patient-reported feedback was collected in the outpatient setting, were related to the higher response rates [44] Social aspects such as access to the Internet or using a patient portal can make the difference as well [8,44,45]. Furthermore, the rates of participation in the surveys and response rates increase when they are promoted by healthcare workers [46].

The completion of both PROM and PREM questionnaires did not require much time, taking around three minutes to complete each. It took longer time for older patients to complete the questionnaires, which may reflect worse technology skills compared to younger patients, as well as aging itself having an impact on completion time [42,47]. Importantly, in cases when the patients completed the forms, in most of the cases (79%) they completed both PROM and PREM. Furthermore, the original PREM instrument was fully completed by all 94 patients, indicating that the questionnaire appeared to the patients as reasonable and user-friendly.

Finally, several limitations have to be addressed regarding the design and conduct of our study. We were unable to collect information about the reasons for non-response of automated surveys, which leads to potential selection bias. We also did not collect the reasons for preferring paper administration over electronic filling of the questionnaire, limiting the understanding of barriers for implementation. Our aim was to develop an instrument that makes it possible to collect patient-reported experiences throughout a whole patient journey at a single time point. Such an approach improves the feasibility of data collection at a wider scale and makes it possible to evaluate the association between past experiences and current self-reported health status. However, longitudinal tracking of PREMs throughout the patient journey at each time point could provide additional important information. In addition, beyond the evaluation of internal consistency, our PREM questionnaire did not undergo further validation. This was primarily due to the pilot nature of our study, the aim of which was primarily to develop a process for creating and implementing an integrated instrument, as well as to analyze the feasibility of such an approach. We did not evaluate the impact of implementing such instrument, as well as did not evaluate the users’ (patients, physicians) feedback and reflections on the instrument and its use. Our study was limited to patients with kidney and hematological diseases from a single tertiary center, thus limiting the results’ generalizability to other health systems and potentially other populations. Finally, our process may not be suitable for institutions or systems that use different EMRs due to limitations related to interoperability and abilities to perform aggregated data analytics.

## 5. Conclusions

In conclusion, we have described a process of developing and implementing an instrument that allows for an integrated collection of PROMs and PREMs for patients requiring long-term management by the example of patients with kidney and hematological diseases. We describe a process of integrating such instrument into the hospital’s information system employing routinely used software tools. Our data indicate the feasibility of such approach, including patient-friendliness (low completion time, remote filling) and ability to provide multidimensional real-time analytics. We also showed that patient-reported experiences throughout all patient journey collected at a single time point are associated with different dimensions of subjective health status at the time of questionnaire completion. This supports the relevance of employing the approach at a wider scale when feasibility aspects become a significant concern. However, a comparative study tracking the experiences longitudinally would be required to identify potential qualitative disparities. Although our approach appears to be feasible, we have identified significant potential barriers: low response rate for e-questionnaires and priority for paper administration among older populations. Thus, studies exploring interventions to improve response rates and tailoring administration modes to different patient populations are needed. Further studies exploring the integration of such instrument into the e-Health system at the national level are needed to assess the reproducibility of our observations.

## Figures and Tables

**Figure 1 healthcare-11-02290-f001:**
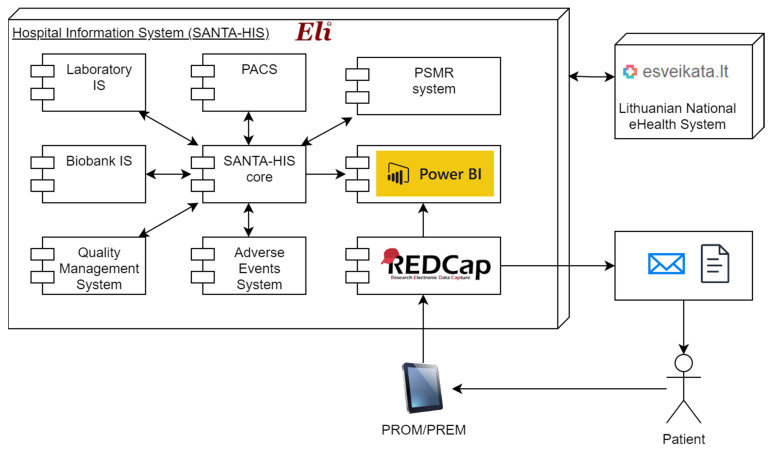
Integrated health information system architecture.

**Figure 2 healthcare-11-02290-f002:**
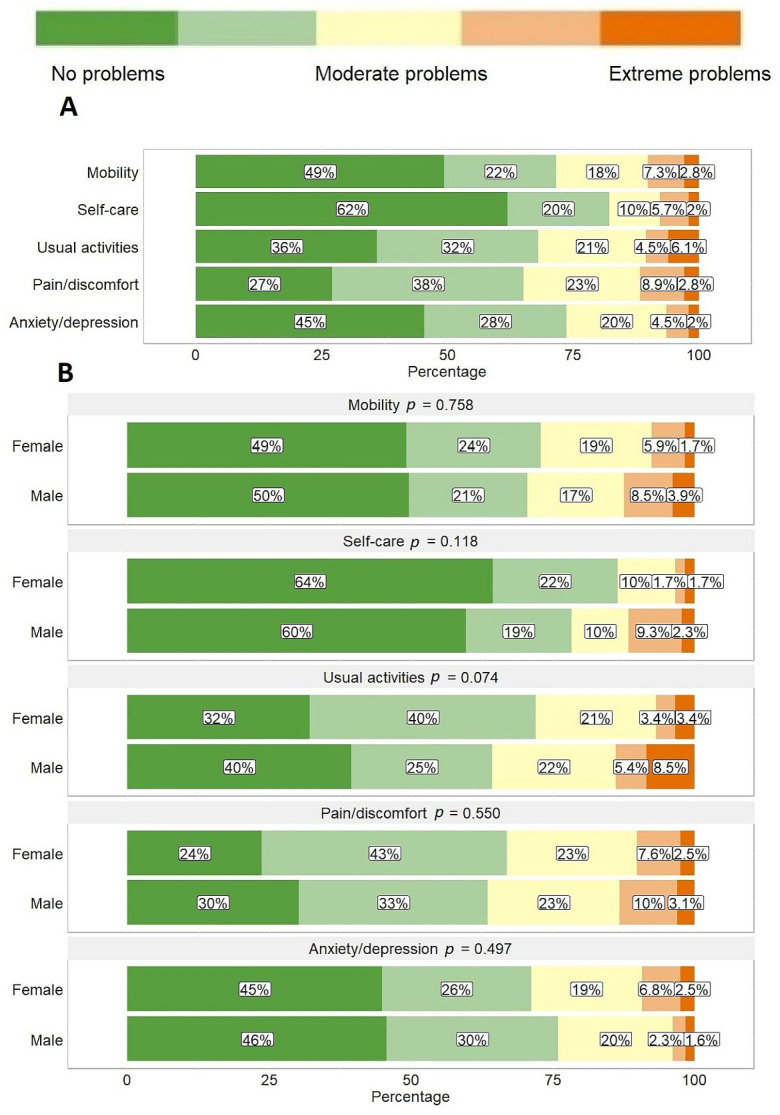
Patient-reported outcome measure (PROM) responses using generic instrument EQ-5D-5L. (**A**) General overview; (**B**) Distribution of responses for females and males. Full questionnaire consisting of 12 questions from Q1 to Q12 is provided as Appendix A.

**Figure 3 healthcare-11-02290-f003:**
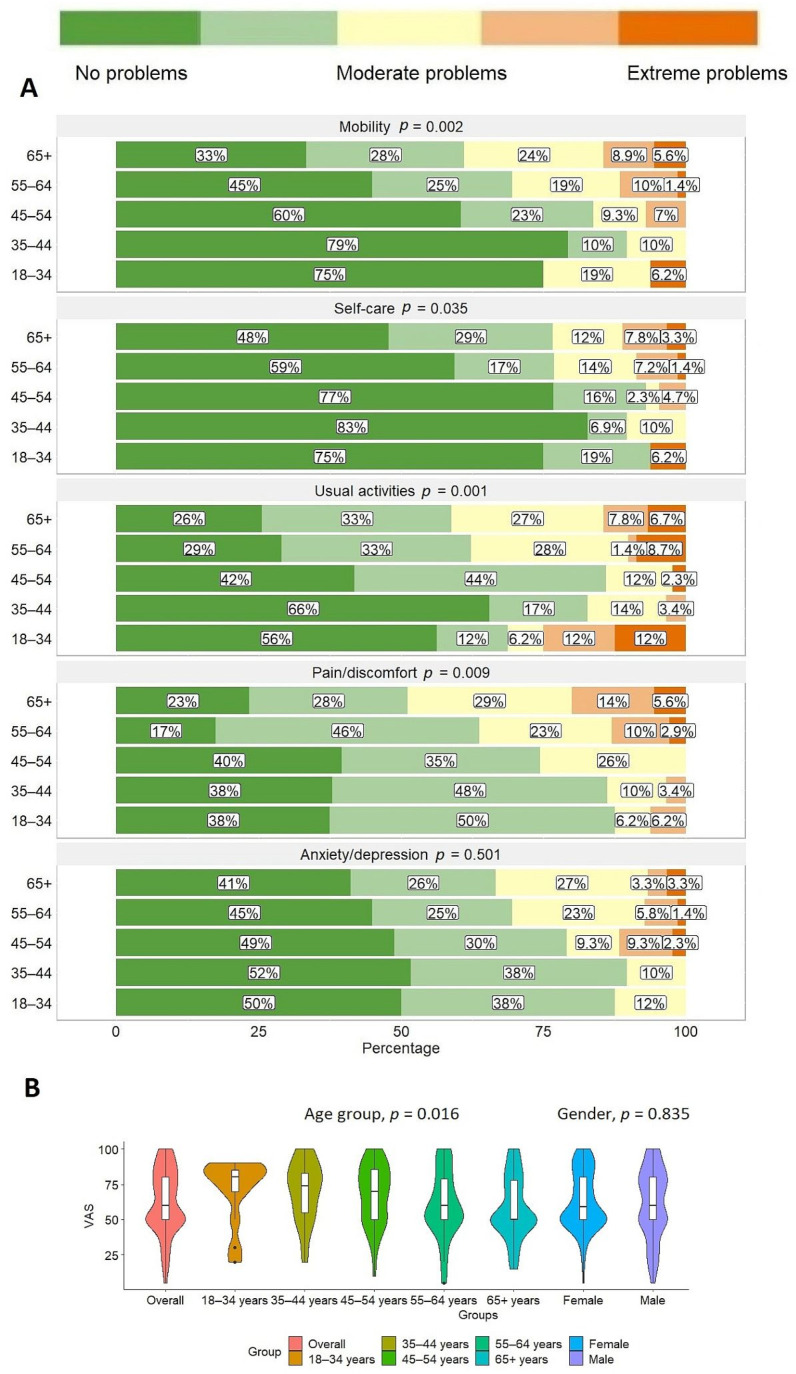
Patient-reported outcome measure (PROM) responses using generic instrument EQ-5D-5L. (**A**) Distribution of responses for different age groups; (**B**) 100-point visual analogue scale (VAS). Full questionnaire consisting of 12 questions from Q1 to Q12 is provided as Appendix A.

**Figure 4 healthcare-11-02290-f004:**
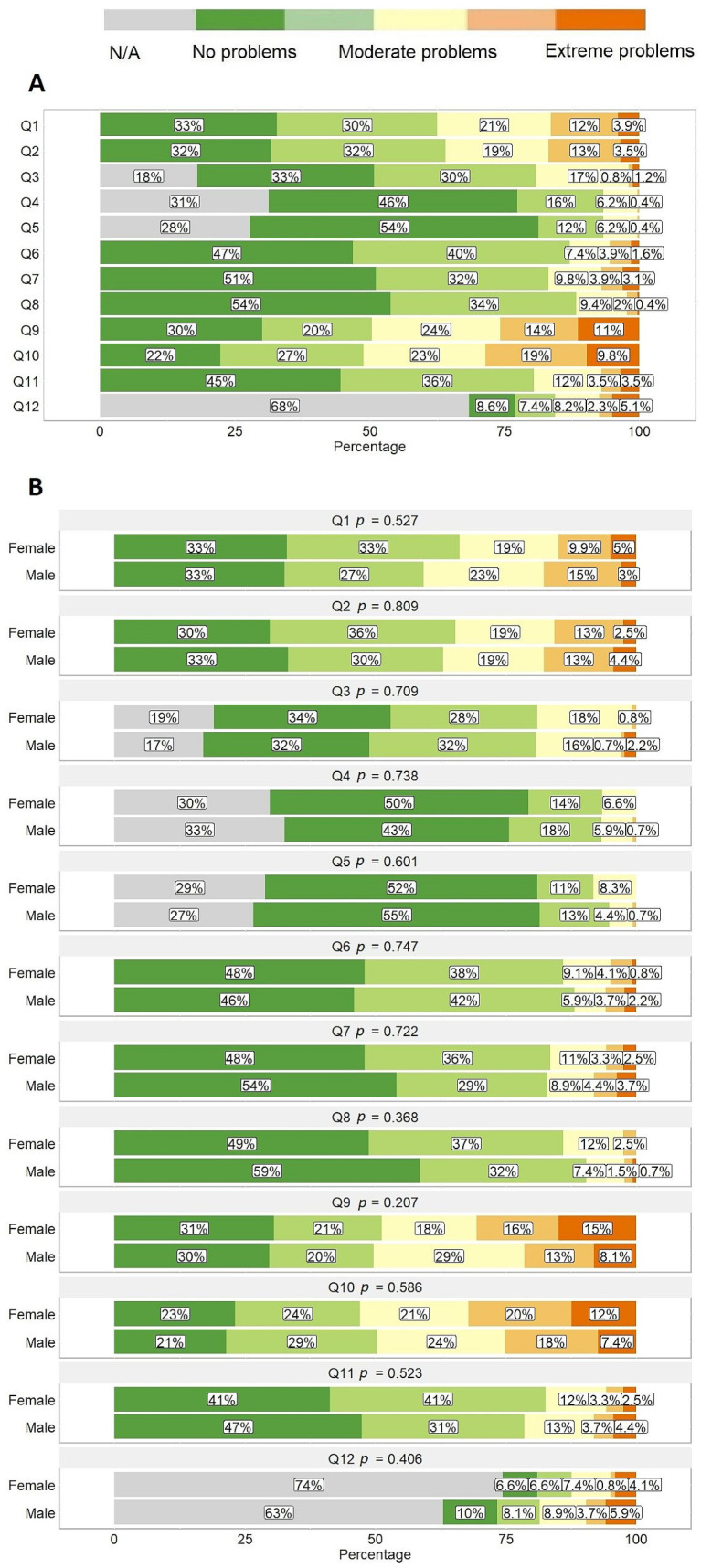
Patient-reported experience measure (PREM) responses using original instrument developed by the study group. Full questionnaire consisting of 12 questions from Q1 to Q12 is provided as Appendix A. (**A**) General overview, (**B**) Distribution of responses for females and males.

**Figure 5 healthcare-11-02290-f005:**
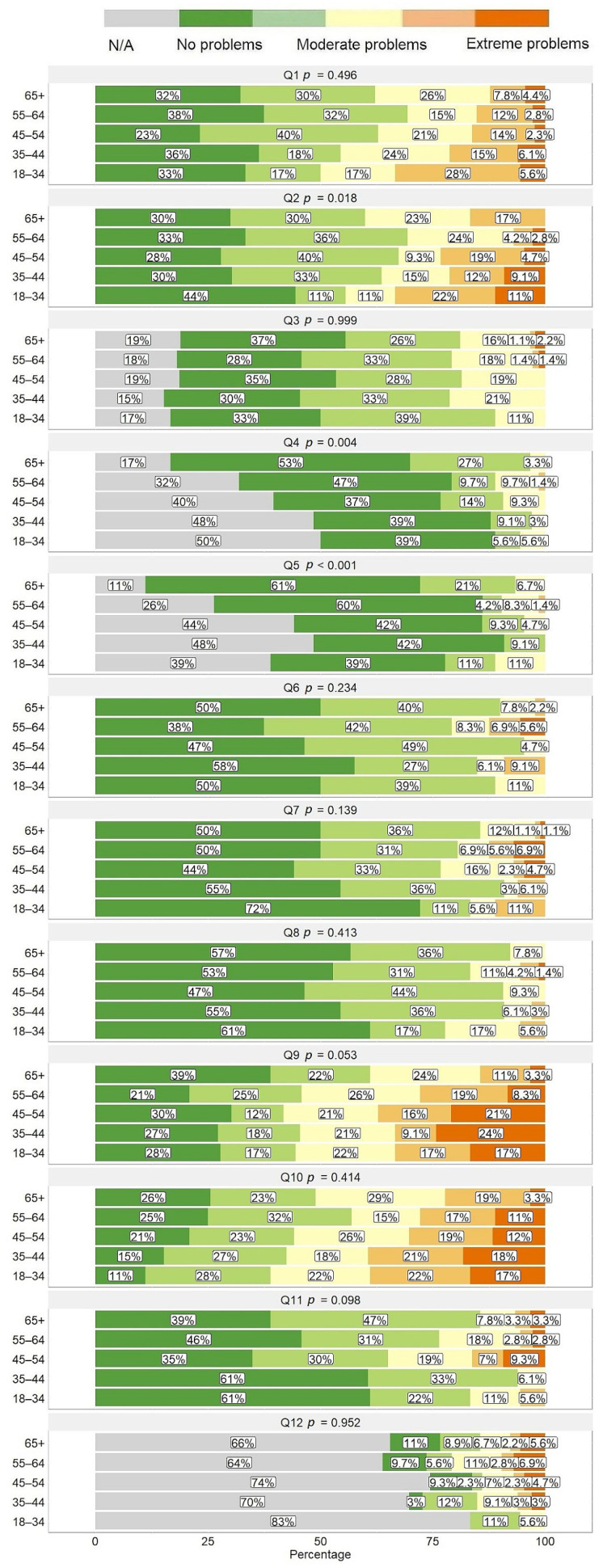
Distribution of patient-reported experience measure (PREM) responses using original instrument developed by the study group for different age group patients. Full questionnaire consisting of 12 questions from Q1 to Q12 is provided as Appendix A.

**Figure 6 healthcare-11-02290-f006:**
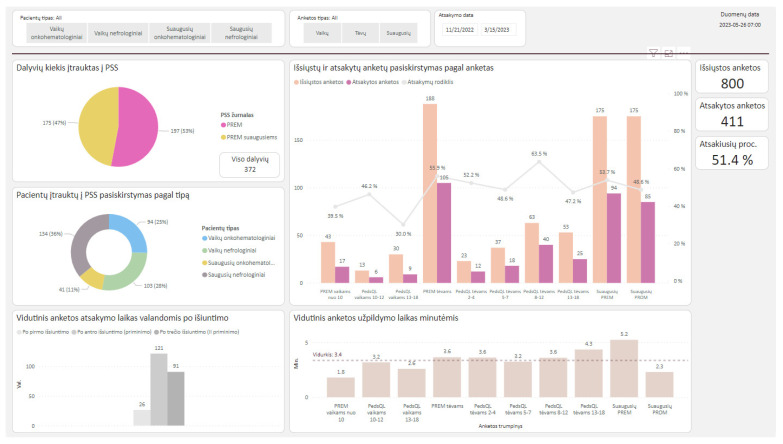
Power BI data visualization.

**Table 1 healthcare-11-02290-t001:** Demographic characteristics of the study population.

Characteristic	N (%)	Age, Median (IQR), Min–Max, Years	*p* Value ^1^
All patients	337	58 (46, 68), 18–90	
Sex			
Female	149 (44.2%)	59 (47, 69), 18–86	0.256
Male	188 (55.8%)	57 (43, 66.3), 19–90	
Disease group			
Kidney	134 (39.8%)	49 (38, 60), 18–87	<0.001
Hematological	203 (60.2%)	63 (52.5, 70), 19–90	
Mode of administration			
Electronic	175 (51.9%)	50 (39.5, 61), 18–88	<0.001
Paper	162 (48.1%)	64 (55, 72), 19–90	

^1^—Wilcoxon rank sum test, IQR—interquartile range.

**Table 2 healthcare-11-02290-t002:** Response rate and completion time for electronic PROM and PREM questionnaires.

		Response Rate, N (%)				Completion Time, Minutes					
Groups	N	PROM		PREM		PROM			PREM		
		N (%)	*p* ^1^	N (%)	*p* ^1^	N ^2^	Median (IQR)	*p* ^3^	N ^4^	Median (IQR)	*p* ^3^
Total	175	85 (48.6)	-	94 (53.7)	-	84	2 (1.3, 2.8)	-	93	3.7 (2.6, 5.3)	-
Sex											
Male	108	49 (45.4)	0.282	55 (50.9)	0.348	48	2.1 (1.4, 3.1)	0.327	54	3.7 (2.5, 5.3)	0.981
Female	67	36 (53.7)		39 (58.2)		36	2 (1.2, 2.5)		39	3.7 (2.6, 5.2)	
Age group											
18–34	25	9 (36)	0.753	11 (44)	0.679	9	1.3 (1.1, 2)	<0.001	11	3.6 (2.1, 4.8)	0.042
35–44	32	16 (50)		20 (62.5)		15	1.1 (0.8, 1.7)		20	2.6 (1.9, 3.4)	
45–54	46	24 (52.2)		24 (52.2)		24	2.2 (1.3, 2.5)		24	4.3 (2.9, 6)	
55–64	44	22 (50)		25 (56.8)		22	2.6 (1.9, 4.7)		25	3.9 (2.8, 5)	
65+	28	14 (50)		14 (50)		14	2.4 (1.9, 2.8)		13	4.0 (3.1, 6.2)	

^1^—Pearson’s Chi-squared test, ^2^—one measurement was removed as an outlier (>100 min), ^3^—Wilcoxon rank sum test, ^4^—one measurement was removed as an outlier (>100 min). IQR—interquartile range; PREM—patient-reported experience measures; PROM—patient-reported outcome measures.

**Table 3 healthcare-11-02290-t003:** Cronbach’s Alpha coefficients for patient-reported measures, PROM and PREM, in overall patients and patient subgroups.

PROM	Cronbach’s Alpha	95% Confidence Interval	
		Lower	Upper
Total	0.869	0.857	0.881
Type of disease			
Nephrological	0.861	0.837	0.882
Hematological	0.865	0.851	0.879
Type of questionnaire			
Electronical	0.874	0.854	0.892
Paper/printed	0.863	0.847	0.877
Sex			
Male	0.898	0.885	0.91
Female	0.825	0.801	0.846
Age group			
18–34	0.784	0.699	0.851
35–44	0.823	0.772	0.864
45–54	0.879	0.851	0.903
55–64	0.892	0.872	0.909
65+	0.843	0.819	0.865
PREM			
Total	0.886	0.872	0.899
Type of disease			
Nephrological	0.831	0.773	0.88
Hematological	0.889	0.874	0.903
Type of questionnaire			
Electronical	0.816	0.764	0.861
Paper/printed	0.891	0.876	0.905
Sex			
Male	0.899	0.883	0.913
Female	0.863	0.834	0.889
Age group			
18–34	0.757	0.584	0.879
35–44	0.673	0.518	0.795
45–54	0.917	0.882	0.945
55–64	0.929	0.913	0.943
65+	0.85	0.822	0.876

## Data Availability

The data presented in this study are available on request from the corresponding author.

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
