# Peer review of "Development and Integration of Patient-Reported Measures into E-Health System: Pilot Feasibility Study"

_healthcare, 2023, doi:10.3390/healthcare11162290_

Round 1

Reviewer 1 Report

The manuscript entitled “Development and Integration of Patient Reported Measures Into E-Health System: Pilot Feasibility Study” was interesting and well-written. The following comments can help the authors to improve it:

1.       Keywords can be selected based on the MeSH terms.

2.       In the introduction section, similar studies and the existing literature related to the integration of patient reported measures into E-Health Systems need to be reviewed.

3.       In the results section, Figures are not clear and need to be replaced with higher resolution Figures.

4.       Research limitations need to be added to the manuscript. Any challenges or limitations can be presented in this section.

5.       Overall, based on the objectives written by the authors, I expected to see “an instrument (including contents, process and integration) for PROM and PREM collection capturing the whole patient journey that could be integrated into the hospital electronic medical records system (EMR). The instrument is presumed to serve as a prototype to be integrated in the existing national e-Health system for continuous monitoring of patient reported measures. Secondary aim: To test the feasibility of the instrument with the group of patients that require long-term management by the specialists in all three levels of the national (Lithuanian) healthcare system”. However, based on my understanding and the content of the manuscript, the instruments were only electronic questionnaires which were completed only once by patients. So, in my opinion, it was not a very specific instrument and couldn’t capture the whole patient journey. Particularly, hematological diseases or chronic kidney diseases (CKD) need continuous attention regarding the patient-reported outcomes. In fact collecting these data is important over time not just once and integrating these data into E-health systems is also important in terms of technical aspects. The type of E-Health system was not clear for me and I have no idea how such a system will respond over time by collecting a huge amount of data for several diseases.

Author Response

Comment #1

Keywords can be selected based on the MeSH terms

The keywords were reviewed and corrected based on MeSH terms.

Comment #2

In the introduction section, similar studies and the existing literature related to the integration of patient reported measures into E-Health Systems need to be reviewed.

We acknowledge that the Introduction section required significant revisions as also pointed out by other Reviewers in order to better reflect current state and challenges of implementing PROMs/PREMs at meso-/macro-levels, as well as also to highlight the scope and relevance of our study. We have significantly revised and changed the Introduction section accordingly.

Comment #3

In the results section, Figures are not clear and need to be replaced with higher resolution Figures.

We thank the Reviewer for this important notion. The figures are now replaced with higher resolution images. For the convenience of the readers we divided Figure 2 into two parts as Figure 2 and Figure 3, therefore, the manuscript now includes six figures instead of five.

Comment #4

Research limitations need to be added to the manuscript. Any challenges or limitations can be presented in this section.

We thank the Reviewer for pointing out the missing Limitations section. It has now been added as a paragraph prior to the Conclusions section.

Comment #5

Overall, based on the objectives written by the authors, I expected to see “an instrument (including contents, process and integration) for PROM and PREM collection capturing the whole patient journey that could be integrated into the hospital electronic medical records system (EMR). The instrument is presumed to serve as a prototype to be integrated in the existing national e-Health system for continuous monitoring of patient reported measures. Secondary aim: To test the feasibility of the instrument with the group of patients that require long-term management by the specialists in all three levels of the national (Lithuanian) healthcare system”. However, based on my understanding and the content of the manuscript, the instruments were only electronic questionnaires which were completed only once by patients. So, in my opinion, it was not a very specific instrument and couldn’t capture the whole patient journey. Particularly, hematological diseases or chronic kidney diseases (CKD) need continuous attention regarding the patient-reported outcomes. In fact collecting these data is important over time not just once and integrating these data into E-health systems is also important in terms of technical aspects. The type of E-Health system was not clear for me and I have no idea how such a system will respond over time by collecting a huge amount of data for several diseases.

We follow the Reviewer’s comment and indeed agree that our instrument is not designed to longitudinally track patient-reported measures, thus providing information on experiences at each stage of patient journey at the time of their occurrence. We also agree with the Reviewer that longitudinal collection of such data would be challenging due to the vast amount of data. Recognizing the challenges related to longitudinal collection we aimed to develop an instrument that would inform health policy-makers and managers on patient-experiences at different stages of patient journey at a single-time point. Moreover, we were aiming to see whether these would correlate with patient-reported outcomes, as has been found in our preliminary results. Patients with kidney and hematological diseases were selected only as representative populations of patients with conditions requiring long-term management to perform the pilot study. The instrument itself, however, is expected to be used for a wider range of conditions requiring similar care pathways.

We thank the Reviewer for identifying that this information was not clearly presented and highlighted throughout the article, thus changes have been made in the Introduction, Methods and Discussion (first paragraph, limitations, conclusions) sections to highlight the aims and to address limitations.

Reviewer 2 Report

The authors have attempted to show the relevance of Patient-reported outcome (PROM) and experience (PREM) measures to deliver quality healthcare. The manuscript is well-designed and articulated. The manuscript is very relevant to understand how the patients understand their condition and also their experience with the system. 

A few minor issues:

Methods: The authors claim the patients were invited for an anonymous survey, which cannot be true as the survey is linked to email addresses and entered in Redcap where they will be identified. Plus there is a considerable number of survey respondents who have filled out a paper-based survey and later entered by the investigators, which will be no longer anonymous. 

The manuscript is well written with detailed description

Author Response

Methods: The authors claim the patients were invited for an anonymous survey, which cannot be true as the survey is linked to email addresses and entered in Redcap where they will be identified. Plus there is a considerable number of survey respondents who have filled out a paper-based survey and later entered by the investigators, which will be no longer anonymous. 

We thank the Reviewer for the comment pointing an important anonymity issue. In our study, e-mail addresses and paper-based questionnaires were processed and placed into the hospital Electronic Medical Records system (ELI) by the data manager, whereas further process of data analysis was fully automated and performed by data analytics system Microsoft Power BI Report Server (Power BI). Therefore, no personal data were accessed or processed by the clinical investigators who were also only able to access aggregated anonymous data for analysis. Appropriate corrections of inaccuracies in methods description were made in the Methods section.

Reviewer 3 Report

Manuscript on a very interesting and important topic. The work needs improvement. In the abstract, I recommend removing the Background, Aim, Methods, and Results headers. In the introduction, I recommend highlighting the gap that the work fills and what is novel about the work compared to other studies. In the methodology, I recommend supplementing the information on how the survey was built, from which parts, and how the respondents were selected for the survey. The work lacks a conclusion section, which I recommend creating and analyzing in terms of the purpose of the work, the methodology used, the results obtained, and achievements in the subject in the literature. Figures 3 and 4 are very interesting but not very clear. It may be worth enlarging them and attaching them to the work as an attachment.

The work contains grammatical and punctuation errors.

The work contains grammatical and punctuation errors.

Author Response

Comment #1.

In the abstract, I recommend removing the Background, Aim, Methods, and Results headers.

Thank you very much for pointing the inaccuracies in the abstract presentation. The abstract has now been corrected according to the requirements.

Comment #2.

In the introduction, I recommend highlighting the gap that the work fills and what is novel about the work compared to other studies.

We acknowledge that the Introduction section required significant revisions as also pointed out by other Reviewers in order to better reflect current state and challenges of implementing PROMs/PREMs at meso-/macro-levels, as well as also to highlight the scope and relevance of our study. We have significantly revised and changed the Introduction section accordingly.

Comment #3

In the methodology, I recommend supplementing the information on how the survey was built, from which parts, and how the respondents were selected for the survey.

We thank the Reviewer for identifying these gaps in the description of our methodology. Accordingly, we have supplemented the Methods section with information on how and where the patients were selected (i.e. consecutive patients meeting eligibility criteria and seen in the day-care, inpatient and outpatient clinics were invited to participate in the study).

Comment #4

The work lacks a conclusion section, which I recommend creating and analyzing in terms of the purpose of the work, the methodology used, the results obtained, and achievements in the subject in the literature.

We agree that conclusion section may strengthen our manuscript and highlight most important points. A Conclusion section describing the aims, primary findings and their relevance, particularly for future research, as well as remaining gaps has been now added.

Comment #5

Figure 3 and 4 are very interesting but not very clear. It may be worth enlarging them and attaching them to the work as an attachment.

We thank the Reviewer for this important notion that has also been pointed out by other Reviewer. The figures are now replaced with higher resolution images. For the convenience of the readers we divided Figure 2 into two parts as Figure 2 and Figure 3, therefore, the manuscript now includes six figures instead of five.

Comment #6

The work contains grammatical and punctuation errors.

The final version was extensively reviewed for English language and punctuation issues.

Reviewer 4 Report

  1. Introduction: The introduction provides a brief overview of the study's background and objectives. However, it would benefit from further contextualization and justification for the importance of patient involvement and patient-oriented decisions in healthcare management strategies. Consider expanding on the current state of patient-centered care and the need for instruments that capture patient-reported outcome and experience measures.

  2. Methods: it lacks specific details about the development process, validation procedures, and how the instrument was integrated into the hospital information and quality management system. Providing more information about the instrument's design, validation methods, and integration process would enhance the clarity and reliability of the study.

  3. Results:  it would be beneficial to provide more specific details regarding the results, such as key findings or trends observed in the pilot study. Consider including a brief summary of the areas of concern identified through the instrument, highlighting their implications for patient-centered care.

  4. Discussion and Conclusion: Consider adding a paragraph to summarize the main findings and their implications. Additionally, provide insights into the potential applications of the instrument and how it can contribute to improving patient-centered care and healthcare quality.

  5. Generalizability: Given that the pilot study focused on patients with hematological or chronic kidney diseases, it is important to acknowledge the potential limitations in terms of generalizability to other patient populations. Consider adding a statement about the potential applicability and transferability of the instrument to different healthcare settings and patient groups.

  6. Overall Structure and Length: The abstract could benefit from a clearer structure that highlights the key aspects of the study, such as objectives, methods, results, discussion, and conclusion. Additionally, ensure that the abstract is concise and within the recommended word limit.

The quality of English language is generally good. The sentences are structured well, and the ideas are effectively communicated. The use of vocabulary and terminology related to the subject matter is appropriate and demonstrates a good understanding of the topic. However, there are a few areas where improvements can be made to enhance the overall quality of the English language:

  1. Sentence Structure: While most sentences are clear and concise, a few sentences could benefit from further clarity and simplification. Consider breaking down longer sentences into shorter ones to improve readability.

  2. Grammar and Punctuation: Pay attention to correct grammar and punctuation throughout the text. There are a few instances where minor errors or inconsistencies are present. Proofreading the text carefully can help identify and correct these issues.

  3. Verb Tenses: Ensure consistency in the use of verb tenses. There are a few instances where the shift between tenses is not clear or consistent. Reviewing the text for verb tense consistency will improve the overall flow of the writing.

  4. Word Choice: While the vocabulary used is generally appropriate, there may be opportunities to vary the word choice and incorporate more descriptive or precise language where applicable. This can help add depth and richness to the writing.

Author Response

Comment #1

Introduction: The introduction provides a brief overview of the study's background and objectives. However, it would benefit from further contextualization and justification for the importance of patient involvement and patient-oriented decisions in healthcare management strategies. Consider expanding on the current state of patient-centered care and the need for instruments that capture patient-reported outcome and experience measures.

We acknowledge that the Introduction section required significant revisions as also pointed out by other Reviewers in order to better reflect current state and challenges of implementing PROMs/PREMs at meso-/macro-levels, as well as also to highlight the scope and relevance of our study. We have significantly revised and changed the Introduction section accordingly, aiming to bring our study into the context and to justify the approach chosen to be studied.

Comment #2

Methods: it lacks specific details about the development process, validation procedures, and how the instrument was integrated into the hospital information and quality management system. Providing more information about the instrument's design, validation methods, and integration process would enhance the clarity and reliability of the study.

We have reviewed the methods section and have added additional details about the development of the custom PREM questionnaire as also suggested by the other Reviewer.

Beyond the evaluation of internal consistency our PREM questionnaire did not undergo further validation. This was primarily due to the pilot nature of our study, the aim of which was primarily to develop a process for creating and implementing such integrated instrument, as well as to analyze the feasibility of such approach. Recognizing this as a limitation of our study, we have added this information to the newly added Limitations section.

Finally, we have also corrected the Methods section by moving some of the information from the Methods to the Integration section and provided additional details about the integration of the instrument into the hospital’s information and quality management system.

Comment #3

Results:  it would be beneficial to provide more specific details regarding the results, such as key findings or trends observed in the pilot study. Consider including a brief summary of the areas of concern identified through the instrument, highlighting their implications for patient-centered care.

We thank the Reviewer for pointing this out. To additionally highlight the areas with the worst patient reported experiences, we have added this information to the introduction of Discussion and the Conclusions sections:

“Our study also allowed to identify areas identified as most problematic through the perspective of patient experiences, namely: psychological and social support, and performance of primary care sector in the diagnosis process and continuous patient management following it.”

“Finally, we have identified several dimensions of patient experiences that were associated with worst evaluations: psychological and social support, and performance of primary care sector in the diagnosis process and continuous patient management following it. Considering the strong associations between patient-reported experiences and subjective health outcomes, this suggests the need of studies evaluating interventions targeting these experiences at the healthcare provider or system level and their effect on patient-reported outcomes.”

Comment #4

Discussion and Conclusion: Consider adding a paragraph to summarize the main findings and their implications. Additionally, provide insights into the potential applications of the instrument and how it can contribute to improving patient-centered care and healthcare quality.

As also pointed out by other Reviewer, we have added a Conclusions section summarizing main findings and their relevance for future research, as well as the remaining gaps in knowledge.

Comment #5

Generalizability: Given that the pilot study focused on patients with hematological or chronic kidney diseases, it is important to acknowledge the potential limitations in terms of generalizability to other patient populations. Consider adding a statement about the potential applicability and transferability of the instrument to different healthcare settings and patient groups.

The information has been added to the Limitations section.

Comment #6

Overall Structure and Length: The abstract could benefit from a clearer structure that highlights the key aspects of the study, such as objectives, methods, results, discussion, and conclusion. Additionally, ensure that the abstract is concise and within the recommended word limit.

Thank you for your valuable comment. The abstract was restructured as per Healthcare Instructions for Authors as well as the aspects of the study were highlighted.  

Comment #7

The quality of English language is generally good. The sentences are structured well, and the ideas are effectively communicated. The use of vocabulary and terminology related to the subject matter is appropriate and demonstrates a good understanding of the topic. However, there are a few areas where improvements can be made to enhance the overall quality of the English language:

  1. Sentence Structure: While most sentences are clear and concise, a few sentences could benefit from further clarity and simplification. Consider breaking down longer sentences into shorter ones to improve readability.
  2. Grammar and Punctuation: Pay attention to correct grammar and punctuation throughout the text. There are a few instances where minor errors or inconsistencies are present. Proofreading the text carefully can help identify and correct these issues.
  3. Verb Tenses: Ensure consistency in the use of verb tenses. There are a few instances where the shift between tenses is not clear or consistent. Reviewing the text for verb tense consistency will improve the overall flow of the writing.
  4. Word Choice: While the vocabulary used is generally appropriate, there may be opportunities to vary the word choice and incorporate more descriptive or precise language where applicable. This can help add depth and richness to the writing.

The manuscript was extensively reviewed and proofread with special attention to grammar, punctuation, sentence structure and word choice.

Round 2

Reviewer 1 Report

I appreciate the authors for their time and efforts to revise the manuscript. Please re-check the manuscript to ensure that there is no mistakes, repetitions, etc. I think "ethical approval" has been repeated twice.

Author Response

We thank the Reviewers for their valuable comments helping to improve our work and the positive feedback after revision. The manuscript underwent final revision and minor remaining issues have been identified and corrected.

Reviewer 3 Report

Thank you for improving the manuscript and for your work.

Author Response

(The authors gave the same response as above.)
